.

# xTrimoABFold: Improving Antibody Structure Prediction without Multiple Sequence Alignments

## Abstract

Antibody, used by the immune system to identify and neutralize foreign objects such as pathogenic bacteria and viruses, plays an important role in immune system. In the field of drug engineering, the essential task is designing a novel antibody to make sure its paratope (substructures in the antibody) binds to the epitope of the specific antigen with high precision. Also, understanding the structure of antibody and its paratope can facilitate a mechanistic understanding of the function. Therefore, antibody structure prediction has always been a highly valuable problem for drug discovery. AlphaFold2, a breakthrough in the field of structural biology, provides a feasible solution to predict protein structure based on protein sequences and computationally expensive coevolutionary multiple sequence alignments (MSAs). However, the computational efficiency and undesirable prediction accuracy on antibody, especially on the complementarity-determining regions (CDRs) of antibody limit its applications on the industrially high-throughput drug design. In this paper, we present a novel method named xTrimoABFold to predict antibody structure from antibody sequence based on a pretrained antibody language model (ALM) as well as homologous templates, which are searched from protein database (PDB) via fast and cheap algorithms. xTrimoABFold outperforms the MSA-based AlphaFold2 and the protein language model based SOTAs, e.g., OmegaFold, HelixFold-Single and IgFold with a large significant margin (30+% improvement on RMSD) while performs 151x faster than AlphaFold2. To the best of our knowledge, xTrimoABFold is the best antibody structure predictor to date in the world.

## 1 Introduction

Antibody is an important type of proteins for disease diagnosis and treatment. The structures of antibodies are closely related to their functions, so that antibody structure prediction, which aims to predicting the 3D coordinates of atoms in antibody, is essential in the biological and medical applications such as protein engineering, modifying the antigen binding affinity, and identifying an epitope of specific antibody. However, manual experimental methods such as X-ray crystallography are time-consuming and expensive.

Recently, deep learning methods have achieved great success in protein structure prediction (Jumper et al., 2021; Baek et al., 2021; Li et al., 2022). In short, these methods incorporate evolutional and geometric information of protein structures and deep neural networks. In particular, AlphaFold2 (Jumper et al., 2021) introduces the architecture to jointly model the multiple sequence alignments (MSAs) and pairwise information, which is able to end-to-end predict the protein structures in near experimental accuracy. Nevertheless, unlike general proteins, antibodies do not evolve naturally but rather they bind to specific antigens and evolve specifically (fast and one-way evolving), the MSAs of antibodies especially on complementarity-determining regions (CDRs) are not always available or reliable, which hurts the accuracy of models on antibody data.

With the development of large-scale pretrained language model, many protein language models (Rao et al., 2020; Elnaggar et al., 2022; Rives et al., 2021; Rao et al., 2021; Ruffolo et al., 2021; Ofer et al.,

2021; Wu et al., 2022) have been developed to generate the representation of protein sequence and show promising performance on contact prediction (Iuchi et al., 2021; Rao et al., 2020), functional properties prediction (Meier et al., 2021; Hie et al., 2021) and structure prediction from single sequence (Hong et al., 2022; Wu et al., 2022; Fang et al., 2022; Chowdhury et al., 2021; Ruffolo & Gray, 2022). These single-sequence-based structure predictors typically follows a two-stage framework that i) trains a protein language model (PLM) on large-scale unlabeled protein databases, e.g., UniRef50, UniRef90, BFD or Observed Antibody Space (OAS) database (Olsen et al., 2022a), ii) employs the evoformer variants and structure module variants to predict protein structures from the learned representation from the pretrained PLM. Their experimental results show comparable accuracy with the standard AlphaFold2 and perform much more efficient because of skipping the computationally expensive CPU-based MSA searching stage. Although large-scale PLM show promising results, neither of PLM-based methods outperform AlphaFold2 on both general protein databases and antibody databases.

**Contribution.** In this paper, we focus on one the of the most important problems in the field of drug discovery: antibody structure prediction. We claim that when conducting structure prediction on antibody, the general protein language model (PLM) is not the best optional. In contrast, we employ an antibody language model (ALM) pretrained on the large-scale OAS database and use evoformer and structure modules to learn the antibody structures in an end-to-end fashion. Also, we design fast and cheap template searching algorithms based on two modalities of both sequence and structures. The searched templates help xTrimoABFold learn from a good starting point. We construct two large antibody databases of 19K antibody structure database and 501K protein structure database from RCSB PDB (Berman et al., 2000) [1]. Experimental results show that our xTrimoABFold performs much better than all the latest SOTAs, e.g., AlphaFold2, OmegaFold, HelixFold-Single and IgFold, with a significant margin (30+% improvement on RMSD). To the best of our knowledge, xTrimoABFold is currently the most accurate antibody structure prediction model of the world. We believe such large improvement on antibody prediction from xTrimoABFold will make a substantive impact on drug discovery.

## 2 RELATED WORKS

**Protein & Antibody Structure Prediction.** Protein structure prediction aims to getting the 3D coordinates from an amino acid sequence, which has been an important open research problems for over 50 years (Dill et al., 2008; Anfinsen, 1973). In recent years, deep learning methods have been widely used in protein structure prediction and considerable progress has been made by using the co-evolution information from Multiple Sequence Alignments(MSAs), such like AlphaFold (Senior et al., 2019; 2020), AlphaFold2 (Jumper et al., 2021), OpenFold (Ahdritz et al., 2021) and RoseTTAFold (Baek et al., 2021). However, these methods are time-consuming and strictly dependent on MSAs, which remains a challenge for the structure prediction of orphan proteins with less homologous information or antibody for which MSAs are not always useful on account of a fast evolving nature. Recently, Lin et al. (2022), Fang et al. (2022) and Wu et al. (2022) make protein structure prediction on large protein language models which are no longer dependent on MSAs, which drastically reduce computation time but incur a certain loss of prediction precision. In particular, models like DeepAb (Ruffolo et al., 2022), ABlooper (Abanades et al., 2022) and IgFold (Ruffolo & Gray, 2022) are specifically developed for antibody structure.

**pretrained Language Model on general protein and antibody.**

- **General Protein Language Model (PLM).** Typically, protein language models (Rao et al., 2020; Elnaggar et al., 2022; Rives et al., 2021; Rao et al., 2021; Ruffolo et al., 2021; Ofer et al., 2021; Wu et al., 2022) employ the popular transformer neural architecture variants (Vaswani et al., 2017) and train on different protein databases, such as UniRef50, UniRef90 and BFD, etc. For example, Rao et al. (2021) introduce ESM variants and use axial attention to learn row and column representation from MSA. Elnaggar et al. (2022) train several PLMs with different number of parameters on UniRef and BFD datasets. Lin et al. (2022) extends ESM and proposed ESM-2, which used relative position encoding to capture the intrinsic interaction of amino acids in the sequence. Wu et al. (2022) pro-

---
[1]The two database are splitted by release datetime on January 17th, 2022.

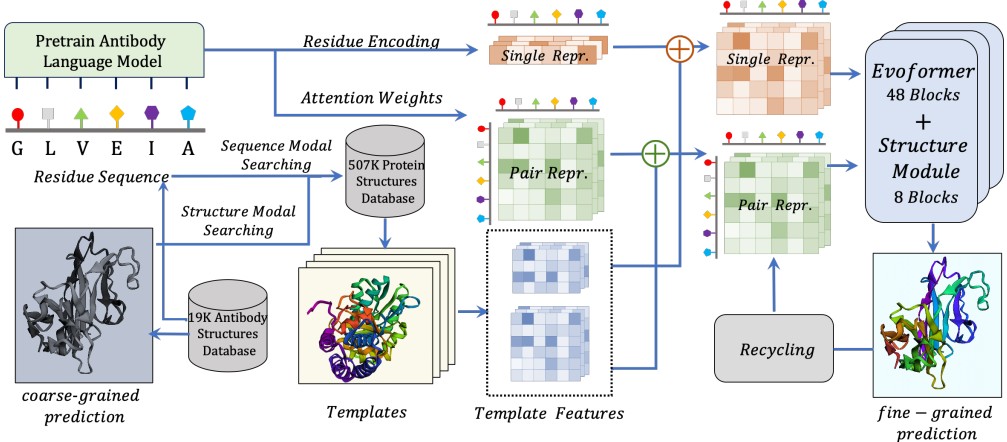

Figure 1: The architecture of xTrimoABFold, which takes single sequence and coarse-grained structural prediction as input, and uses pretrained antibody language model and cross-modal templates to model homologous sequences and structures respectively. Then, the combination of Evoformer (encoder) and Structure Module (decoder) of AlphaFold2 is employed to predict the fine-grained prediction 3D structure of antibody.

pose OmegaPLM and use gated attention module (GAT) to replace self-attention layers in transformer.

- **Antibody Language Model (ALM).** For antibody problems, language model trained on antibody sequences may learn more specific language information and can perform more powerful representations than general PLM. Sapiens (Prihoda et al., 2022) consisting of two Transformer models were trained on 20M BCR heavy chains and 19M BCR light chains. DeepAb is a long short-term memory (LSTM) network which is trained on 100k paired BCR sequences of Observed Antibody Space (OAS) database (Olsen et al., 2022a). AbLang (Olsen et al., 2022b) contains two models for either light chains or heavy chains trained on OAS. AntiBERTa (Choi, 2022) and AntiBERTy Ruffolo et al. (2021) are both BERT-based pretrained language model for antibody trained on OAS with no distinction on light and heavy chains.

**Template Searching Algorithms and Tools** . For protein structure prediction, templates structures can be a kind of auxiliary information to improve the quality of structure models. HH-Search (Steinegger et al., 2019) applied by AlphaFold2 for template searching is an open source tool mainly used to template detection by HMM-HMM alignments between query and target database. Different from HHSearch by making a search based on MSAs, FoldSeek (van Kempen et al., 2022), can support a fast and accurate structure searching on datasets by using a structure as a query.

## 3 METHODOLOGY

In this section, we propose xTrimoABFold, an antibody structure prediction pipeline based on the AlphaFold2 architectures, but without the computationally expensive CPU-based MSA searching. Specifically, xTrimoABFold uses a pretrained antibody language model to generate residue encoding and pair attentions to initialize single and pair representations respectively, which can compensate the loss of homologous information of MSAs. Also, xTrimoABFold adopts a cross-modal template searching algorithm, which can quickly search homologous structures in both sequential and structural modals. The overview of xTrimoABFold is shown in Figure. 1.

### 3.1 SINGLE SEQUENCE MODELING WITH PRETRAINED ANTIBODY LANGUAGE MODEL

The excellent performance on many downstream tasks , e.g., protein structure prediction, drug-target interaction of pretrained protein language models (PLMs) shows that PLMs can mine homologous

sequence information without complex manual preparation of MSAs. Therefore, xTrimoABFold uses a pretrained antibody language model to generate single and pair representations instead of MSAs.

**Single Representation.** By default, pretrained antibody language model generates residue (token) level representations with a single sequence as input, which can be used as the initial single representation of following evoformer (a transformer-based encoder) by proper transformation. Since the language model adopts the mechanism of multi-head self-attention, each token can get information from other tokens, which can be seen as a pair2residue communication. Formally, given a residue sequence of an antibody, the computation of single representation is as follows:

$$\boldsymbol{z} = PLM_{ab}(\boldsymbol{x}), \boldsymbol{s}^0 = Linear(\boldsymbol{z}), \boldsymbol{z} \in \mathbb{R}^{N \times d_{lm}}, \boldsymbol{s}^0 \in \mathbb{R}^{N \times d_s} \tag{1}$$

where $N$ refers to the number of residues in the given protein, $d_{lm}, d_s$ are the hidden sizes of language model and following evoformer respectively, $PLM_{ab}$ is pretrained antibody language model, $\boldsymbol{x} = \{x_1, x_2, \cdots, x_N\}$ denotes the sequence of residues, and $Linear$ refers to the linear layer of neural network. Then, $s^0$ is used as the initial single representation of evoformer.

**Pair Representation.** The attention weights of multi-head self-attention mechanism in antibody language model are rich in prior knowledge about the relation between residues such as position information, which can be naturally combined as the initial pair representation through adaptive transformation. Specifically, the generation of initial pair representation can be formalized as follows:

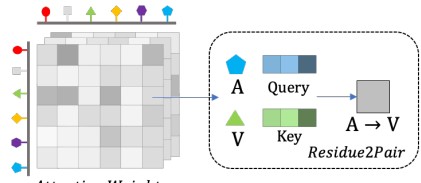

Figure 2: An illustration of the residue2pair communication.

$$\boldsymbol{q}_{ij} = \text{Concat}\left(\left\{\text{softmax}\left(\frac{\boldsymbol{Q}_i^{k,l}(\boldsymbol{K}_j^{k,l})^T + \boldsymbol{a}_{ij}}{\sqrt{d_{lm}}}\right)\Big| l \in [1, L], k \in [1, H]\right\}\right) \tag{2}$$

where $\boldsymbol{p}_{ij}^0 = Linear(\boldsymbol{q}_{ij})$, $\boldsymbol{q} \in \mathbb{R}^{N \times N \times HL}$, $L$ and $H$ denote the number of layers and attention heads respectively, $\boldsymbol{Q}_i^{k,l}$ and $\boldsymbol{K}_j^{k,l}$ are the query and key vectors of residues $i$ and $j$ in $l$-th layer and $k$-th head respectively, $a_{ij}$ denotes the relative position encoding between $i$ and $j$, and $\boldsymbol{p}^0 \in \mathbb{R}^{N \times N \times d_p}$ is the initial pair representation.

The above calculation can be regarded as residue2pair communication because of multi-head query-key product of residue pairs are involved in this step.

### 3.2 STRUCTURE PRIOR: CROSS-MODAL HOMOLOGOUS STRUCTURE SEARCHING

Structure templates typically provide a good prior for structure prediction. The previous works such as AlphaFold2 search templates by MSAs-based algorithms, e.g., MMseqs2 (Steinegger & Söding, 2017), HHBlits (Remmert et al., 2012), which are expensive, inefficient and highly depend on MSAs quality. In this paper, we provide an memory and computation efficient cross-modal template searching algorithm, which introduces two perspectives of sequence and structure to search templates without MSAs. Before conducting template search, we construct a novel antibody database named Antibody Data Bank (ADB), which is derived from RCSB Protein Data Bank (PDB) (Berman et al., 2000) and contains 501K proteins structures.

**Sequential Modal Searching.** Taking into account the idea that similar antibody sequences are likely to have similar 3D structures [2], we use sequence alignment based similarity score to search the structures with similar sequences to target antibody from the database as the templates. The similarity score function can be formalized as:

---

[2]Although this idea is not work everywhere, we believe it could provide a well prior empirically.

$$\text{Sim}(\boldsymbol{x}_1, \boldsymbol{x}_2) = \text{Align}(\boldsymbol{x}_1, \boldsymbol{x}_2)) / \max(\text{len}(\boldsymbol{x}_1), \text{len}(\boldsymbol{x}_2)), \tag{3}$$

where $\boldsymbol{x}_1$ and $\boldsymbol{x}_2$ are residue sequences, and $Align(.,.)$ is the sequence alignment, which denotes the maximum matched residues between two amino acid sequences (e.g., $Align($'GVI','GIV'$) = 2$), we use the Needleman-Wunsch algorithm (Likic, 2008) for sequence alignment computation. Sequential modal searching first screens out all sequences whose similarity scores are in the range of $(0.4, 0.95)$, and restricts the available templates to up to 10 with the maximum similarity scores [3] to target sequence database. After that, these top 10 structures will be considered as a part of template candidates for the following training or inference.

**Structural Modal Searching.** Structural modal searching focuses on finding similar structures in the database based on the coarse-grained structural prediction of the target antibody even though the sequences of these structures may not match the target. We use target sample in AlphaFold Protein Structure Database [4] as the coarse-grained structural prior. Similar to sequential modal searching, we compute the alignment scores between structure of target antibody and structures in database and remove the target itself and the structures with too high similarity to the target structure from database. We add the top 10 structures to the template candidate set. We use FoldSeek (van Kempen et al., 2022), a fast tool suitable for structure-pairwise alignment to calculate alignment scores.

**Template Features** After cross-modal template searching, we can get $T$ template candidates, $T$ is less than or equal to 20 because of the duplication of two modal search results. We choose 4 templates from candidate set of top-2 sequential modal templates and top-2 structural modal templates at inference time. In the training step, we randomly select $min(Uniform[0,T], 4)$ templates out of this restricted set of $T$ templates, and we believe that the structures selected by two searching algorithms contain more homologous structure information, so that we assign higher sampling probability to these structures. We employ the template encoder of AlphaFold2 to encode the template structures into two types of template features of template angle features and template pair features, and add them to the initial single and pair representations respectively, which can be formalized as follows:

$$\hat{\boldsymbol{s}}^0 = \text{Concat}(\boldsymbol{s}^0, \boldsymbol{f}_{ta}), \ \hat{\boldsymbol{p}}^0 = \boldsymbol{p}^0 + \boldsymbol{f}_{tp}; \tag{4}$$

where $\boldsymbol{f}_{ta} \in \mathbb{R}^{4 \times N \times d_s}$, $\hat{\boldsymbol{s}}^0 \in \mathbb{R}^{5 \times N \times d_s}$, $\hat{\boldsymbol{p}}^0, \boldsymbol{f}_{tp} \in \mathbb{R}^{N \times N \times d_p}$, $\boldsymbol{f}_{ta}$ and $\boldsymbol{f}_{tp}$ are the template angle and pair features respectively, and $\hat{\boldsymbol{s}}^0$ and $\hat{\boldsymbol{p}}^0$ are the initial single and pair representations with template features.

### 3.3 XTRIMOABFOLD

**Evoformer and Structure Module.** We use the Evoformer of AlphaFold2 to model complex information in initial single and pair representations. $\hat{\boldsymbol{s}}^0$ and $\hat{\boldsymbol{p}}^0$ can be naturally taken as the input of Evoformer. Note that the column-wise gated self-attention of Evoformer is necessary, which can exchange the sequence information modeled by the antibody language model with the structure information of templates. We follow the other important components of AlphaFold2, $i.e.$, Structure Module, which employs a number of geometric transformation operators such as Invariant Point Attention to predict the 3D structures of the protein end-to-end. Moreover, a recycling mechanism is employed to refine the predicted structures iteratively followed AlphaFold2.

**Training Objective.** xTrimoABFold is trained end-to-end by a variation of the loss function proposed by AlphaFold2 of Frame Aligned Point Error and a number of auxiliary losses, which can be formalized as follows:

$$\mathcal{L}_{\text{train}} = 0.5\mathcal{L}_{\text{FAPE}} + 0.5\mathcal{L}_{aux} + 0.3\mathcal{L}_{dist} + 0.01\mathcal{L}_{conf}, \tag{5}$$

---

[3]One can tune number of selected templates accordingly.

[4]https://alphafold.ebi.ac.uk/

Table 1: Statistics of datasets

| Dataset | Protein Count | Residue Count | | | | Resolution Statistic | | | |
|---|---|---|---|---|---|---|---|---|---|
| | | min | max | mean | std | min | max | mean | std |
| **Training BCR Set** | 18470 | 71 | 879 | 191 | 46 | 0.92 | 9.00 | 2.82 | 0.95 |
| **Test BCR Set** | 470 | 97 | 236 | 183 | 47 | 1.14 | 7.51 | 2.81 | 0.78 |
| **RCSB PDB Set** | 501533 | 2 | 4433 | 259 | 199 | 0.48 | 9.00 | 2.59 | 1.00 |

where $\mathcal{L}_{\text{FAPE}}$ refers to the framed aligned point error (FAPE) over all atoms (Jumper et al., 2021), $\mathcal{L}_{aux}$ are the averaged FAPE and torsion losses on the inter-mediate structures over $C_\alpha$ only, $\mathcal{L}_{dist}$ is an averaged cross-entropy loss for distogram prediction, $\mathcal{L}_{conf}$ is the model confidence loss, these four loss functions are proposed by AlphaFold2. Compared to AlphaFold2, xTrimoABFold removes the loss on masked MSA, because we don't need MSA anymore.

**Fine-tuning with CDR focal loss** Since the structure of complementarity determining region (CDR) in antibody is usually hard to predict than other framework regions (FR), we fine-tune xTrimoABFold with a CDR focal loss after training. Formally, the CDR focal loss is denoted as:

$$\boldsymbol{x}_{ij} = T_j^{-1} \circ \boldsymbol{x}_i, \ \boldsymbol{x}_{ij}^{\text{true}} = T_j^{\text{true}^{-1}} \circ \boldsymbol{x}_j^{\text{true}}, \ T_j, T_j^{\text{true}} \in (\mathbb{R}^{3\times3}, \mathbb{R}^3), \ \boldsymbol{x}_i, \boldsymbol{x}_i^{\text{true}} \in \mathbb{R}^3. \quad (6)$$

$$d_{ij} = \sqrt{\left\| \boldsymbol{x}_{ij} - \boldsymbol{x}_{ij}^{\text{true}} \right\|^2 + \epsilon}, \ \epsilon = 10^{-4} \text{\AA}^2 \quad (7)$$

$$\mathcal{L}_{\text{fc}_{\text{CDR}}} = \frac{1}{Z} \frac{1}{N_{\text{atoms}}^{\text{CDR}}} \sum_{i \in \{1, \cdots, N_{\text{atoms}}^{\text{CDR}}\}} \frac{1}{N_{\text{frames}}} \sum_{j \in \{1, \cdots, N_{\text{frames}}\}} \min(d_{clamp}, d_{ij}), \ d_{clamp}, Z = 10 \text{\AA} \quad (8)$$

$$\mathcal{L}_{\text{fine-tune}} = \mathcal{L}_{\text{train}} + \lambda \mathcal{L}_{\text{fc}_{\text{CDR}}} \quad (9)$$

where $\boldsymbol{x}_i, \boldsymbol{x}_i^{\text{true}}$ are the prediction and ground-truth 3D coordinates of atom $i$ in CDR regions respectively, $T_j, T_j^{\text{true}}$ are the SE(3) transformations, $N_{\text{atoms}}^{\text{CDR}}$ denotes the number of atoms in CDR regions of antibodies, and $N_{\text{frames}}$ is the number of local frames. Fine-tuning with $\mathcal{L}_{\text{fine-tune}}$ helps xTrimoABFold pay more attention to the difficult CDR regions.

## 4 EXPERIMENTS

### 4.1 EXPERIMENTAL SETUP

**Datasets.** In the experiments, we created two large datasets. The first one is 19K antibody structure dataset. We got a total of 18937 antibody data consisting of both amino acid sequences and structures selected from RCSB Protein Data Bank (PDB) (Berman et al., 2000) released before April 13th, 2022. The specific selections focusing on the structures and sequences are as follows. We firstly split each PDB file into single chains, and then make the selection. On the one hand, among the whole 19736 BCR chains from PDB, samples which have no structure resolution values or those of which the structure resolution is larger than 9Å were filtered out to keep the quality of structure data. On the other hand, as for the sequences, we filtered out the samples whose sequence is empty or those the repetition rate of a kind of amino acid is more than 90 percent in a sequence. Besides, we also conducted deduplication on the sequence and kept the samples which have lower structure resolution. After these filtering processes, we got 18937 antibody data as our dataset. Among these, we collected data released before January 17th, 2022 as the training set which contains 18470 samples and other 470 samples to be the test set. The second dataset is our 501K protein structure database. We got the whole protein database downloaded from RCSB PDB. Filtering out the missing structure file, we got a total of 593491 protein chains. Later, we deleted the parts out of specification on structure resolution and sequence similarity mentioned before. After that, we finally get rid of the repeated examples getting total 501533 protein chains. Table 1 provides detailed statistics.

**Baseline.** We compare our xTrimoABFold method with 4 latest state-of-art protein structure prediction methods: AlphaFold2 (Jumper et al., 2021), PLM-based HelixFold-Single (Fang et al., 2022) and OmegaFold (Wu et al., 2022), ALM-based IgFold (Ruffolo & Gray, 2022). For AlphaFold2, we made the inference using five different models and picked up the structures with highest pLDDT [5] confidence for benchmarking. We also trained a variant of our model called xTrimoABFold-ESM, which replaces the antibody language model to a general protein language model of ESM2 Lin et al. (2022). The performance of xTrimoABFold-ESM is worse than xTrimoABFold, which demonstrates that antibody language model is a better optional than general protein language model.

**Evaluation metrics.** To evaluate the quality of antibody structure prediction, we used root-mean-squared-deviation (RMSD), TM-Score (Zhang & Skolnick, 2004), GDT_TS and GDT_HA Zemla et al. (1999) as the evaluation metric. Both two values are calculated over backbone heavy atoms after alignment of the respective framework residues by DeepAlign (Wang et al., 2013). In order to evaluate the performance on CDR loops which are considered difficult for model to predict (Ruffolo & Gray, 2022), we calculated these two metrics on both antibody structures and CDR loop structures.

$$\text{TM-Score} = \max \left[ \frac{1}{L_{target}} \sum_{i}^{L_{common}} \frac{1}{1 + \left( \frac{d_i}{d_0(L_{target})} \right)^2} \right],$$

where $L_{target}$ is the sequence length of target protein and $L_{common}$ is the number of residues that appear in both the template and target structures.

**Hyperparameter Settings.** We used AntiBERTy (Version 0.0.5, installed from PyPI), a BERT-based pretrained protein language model trained on OAS with 558M antibody natural sequences to generate residue level representations. The hidden dimension of the model is 512 and the feedward dimension is 2048 AntiBERTy contains 8 layers, with 8 attention heads per layer. In total, AntiBERTy contains approximately 26M trainable parameters (Ruffolo et al., 2021). In the training part, we block the gradient backpropagation of antibody language model and just train the Evoformer and Structure Module. We use the Adam Optimizer (Kingma & Ba, 2014) with the learning rate of 1e-3, $\beta_1 = 0.9$, $\beta_2 = 0.999$, $\epsilon = 8$ and weight decay of 0. Moreover, We also clip the gradient using the threshold of 10e9. Our model was trained for 25 epochs in 46 hours on 8 NVIDIA A100 GPUs with a stayed batch size of 8. The same as AlphaFold2, the crop size of sequence is set into 256. On account of the replacing of multi- sequence alignments(MSA) representation with the single sequence representation of antibody language model, we removed *InputEmbedder*, *ExtraMSAEmbedder* and *ExtraMSAStack* as well as the *masked MSA loss* compared to AlphaFold2. When making structural modal searching, Foldseek which enables fast and sensitive comparisons of large structure sets was used. While searching templates from PDB, the following flags were set to a non-default value for this tool. We chose 3Di Gotoh-Smith-Waterman as the alignment type and set max-seq into 2000.

## 4.2 Results of main experiments

In this subsection, we show the main results and compare our xTimoABFold with four baselines. Main results contain two part, one of which is the model performance on evaluation metrics, and another is for the time efficiency. Table 2 and 3 respectively show the accuracy performance of all models on antibody structure prediction and CDR loop structure prediction. For brevity, we only present RMSD and TM-score for 3 CDR loops. Moreover, we present the performance with respect to the antibody structure prediction time of each methods on different length of amino acid sequence from the test dataset in Figure 3.

**Performance on antibody structure.** xTrimoABFold significantly outperforms all baselines on test dataset for antibody structure prediction. In terms of RMSD, xTrimoABFold makes 32.60%, 35.42%, 28.97%, 85.21% improvements over AlphaFold2, OmegaFold, HelixFold-Single and IgFold. In the meanwhile, this trend continues on other evaluation metrics. That is to say, our xTrimoABFold achieves state-of-art performance on the antibody structure prediction compared with

---

[5]pLDDT: predicted local distance difference test.

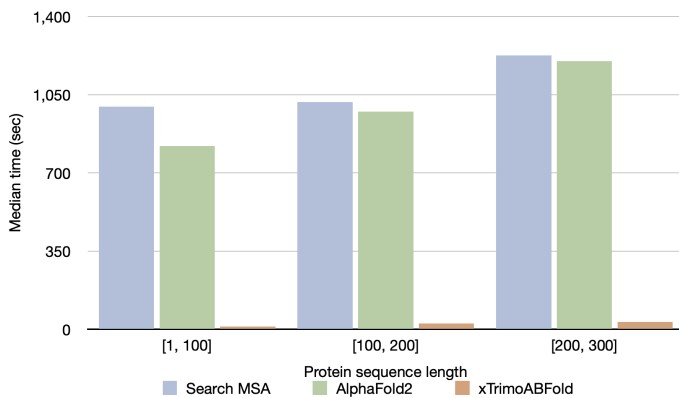

Figure 3: Median time of MSA search, AlphaFold2 and xTrimoABFold. xTrimoABFold is 151 faster than AlphaFold2.

Table 2: Experimental results of antibody structure prediction on test dataset with 95% confidence interval. xTrimoABFold-ESM refers the similar approach like xTrimoABFold expect for replacing the pretrained ALM to the pretrained PLM: ESM2 (Lin et al., 2022) with 15b parameters (the largest PLM to date). The results show ALM is more suitable for antibody structure prediction.

| Method | RMSD ↓ | TM-Score ↑ | GDT_TS ↑ | GDT_HA ↑ |
|---|---|---|---|---|
| AlphaFold2 | 3.1254 ± 0.1410 | 0.8385 ± 0.0055 | 0.7948 ± 0.0057 | 0.6548 ± 0.0063 |
| OmegaFold | 3.2610 ± 0.1463 | 0.8384 ± 0.0057 | 0.7925 ± 0.0059 | 0.6586 ± 0.0063 |
| HelixFold-Single | 2.9648 ± 0.0997 | 0.8328 ± 0.0055 | 0.7805 ± 0.0057 | 0.6225 ± 0.0060 |
| IgFold | 14.2407 ± 0.4115 | 0.5035 ± 0.0143 | 0.4495 ± 0.0157 | 0.3474 ± 0.0148 |
| xTrimoABFold | **2.1059 ± 0.0812** | **0.8936 ± 0.0053** | **0.8655 ± 0.0058** | **0.7439 ± 0.0068** |
| xTrimoABFold-ESM | 4.8790 ± 0.1299 | 0.7057 ± 0.0061 | 0.6727 ± 0.0063 | 0.4992 ± 0.0062 |

not only PLM-based but also MSA-based protein structure prediction methods. To the best of our knowledge, our xTrimoABFold has achieved optimal performance on antibody structure prediction.

**Performance on CDR loops.**    As for the structure prediction of CDR loops, which are well-known as difficult domains for model to make accurate prediction, our xTrimoABFold also performs well. In detail, xTrimoABFold has relatively improvements over HelixFold-Single and IgFold, which are trained based on a large scale protein language model and antibody language model on CDR1 and CDR2 loop. Concretely, we notice that our xTrimoABFold yields the best performance in CDR3 loop which has been proven a difficult domain to predict because of the highly variable and conformationally diverse.

**Time efficiency on structure prediction**    xTrimoABFold performed a fast antibody structure prediction. AlphaFold2 make protein structure prediction according to MSAs, which results in massive time consumption. Compared with AlphaFold2, our xTiomoABFold is a MSA-free model which predict the protein structure by a single amino acid sequence with MSA searching. From Figure 3 we can find that, our model performs 151X faster than AlphaFold2, which overcomes the bottleneck of time efficiency in protein structure prediction by AlphaFold2, enables large-scale antibody structures prediction in a fast speed.

## 4.3 ABLATION STUDY OF MODEL COMPONENTS

In this subsection, we make ablation studies to evaluate the performance improvement brought by our the introducing of pretrained antibody language model Antiberty and the added CDR focal loss when fine-tuning the model.

Table 3: Experimental results of Antibody CDR loop structure prediction on test dataset with 95% confidence interval.

| Method | CDR1 | | CDR2 | | CDR3 | |
|---|---|---|---|---|---|---|
| | RMSD ↓ | TM-Score ↑ | RMSD ↓ | TM-Score ↑ | RMSD ↓ | TM-Score ↑ |
| AlphaFold2 | 0.7151 ± 0.0409 | 0.6396 ± 0.0070 | 0.6733 ± 0.0360 | 0.6669 ± 0.0069 | 1.4794 ± 0.0682 | 0.5075 ± 0.0090 |
| OmegaFold | 0.7202 ± 0.0443 | 0.6457 ± 0.0073 | 0.6804 ± 0.0406 | 0.6750 ± 0.0069 | 1.4861 ± 0.0685 | 0.5114 ± 0.0092 |
| HelixFold-Single | 1.5964 ± 0.1093 | 0.3361 ± 0.0131 | 1.1659 ± 0.0894 | 0.3564 ± 0.0137 | 1.6303 ± 0.1088 | 0.2132 ± 0.0116 |
| IgFold | 2.7411 ± 0.1059 | 0.3138 ± 0.0093 | 2.6774 ± 0.1022 | 0.3339 ± 0.0093 | 3.3222 ± 0.1055 | 0.2104 ± 0.0078 |
| xTrimoABFold | **0.6064 ± 0.0294** | **0.6585 ± 0.0072** | **0.5899 ± 0.0278** | **0.6823 ± 0.0067** | **1.2539 ± 0.0584** | **0.5517 ± 0.0095** |

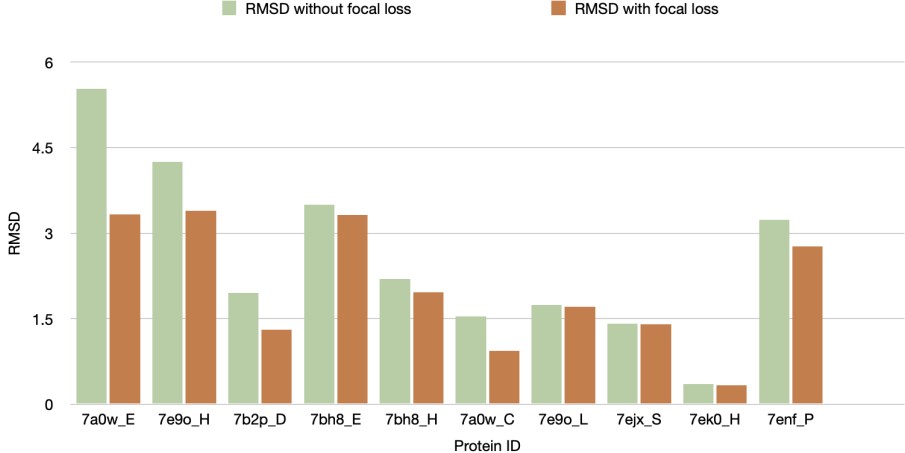

Figure 4: Model performance on RMSD w/o CDR focal loss

**Performance gains from pretrained antibody language model** xTrimoABFold used pretrained antibody language model, AntiBERTy to generate residue-level representations, which contains more specific antibody information compared to general protein language model like OmegaPLM, ESM-2(Rives et al., 2021) etc. Here we fixed the other part of xTrimoFold and use ESM-2, a large-scale protein language model trained on 250 million protein sequences to validate our choice of antibody language model rather than regular protein language model. This model was trained on the same setting of xTrimoABFold and get worse performance compared to xTrimoABFold, the prediction performance can be seen in Table 2

**Performance gains from CDR focal loss** From test dataset, we randomly selected ten samples and compare the improvements of model performance before and after adding CDR focal loss. In these examples, we observed various degree of decrease of RMSD value compared from predicted structure to the ground truth Figure 4, meaning that the our focal loss truly makes sense in the antibody structure prediction especially for the CDR loops which seems difficult to predict for regular models.

## 5 CONCLUSION

In this paper, we have proposed a promising model, xTrimoABFold, for antibody structure prediction. On the one hand, xTrimoABFold employs the pretrained antibody language model to extract the information of single sequence, which is performs well than traditional protein language models. On the other hand, xTrimoABFold uses a efficient template searching algorithm based on two modalities of both sequences and structures.

**Limitations and Broader Impact.** The limitation of this work is that we only address the antibody structure prediction although it's important on drug discovery. In the future, we will extend this work to general protein prediction and the complex prediction.

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
