# OpenReview forum: "xTrimoABFold: Improving Antibody Structure Prediction without Multiple Sequence Alignments "
_ICLR.cc/2023/Conference — Submitted to ICLR 2023_

### Official Review · Reviewer_1hLD · 2022-11-01

**Confidence:** 4
**Correctness:** 4
**Technical Novelty And Significance:** 1
**Empirical Novelty And Significance:** Not applicable
**Recommendation:** 3

**Clarity, Quality, Novelty And Reproducibility:**

The description of this paper is mostly clear but the novelty of this paper is very weak

**Strength And Weaknesses:**

The major strength of this paper is the superior performance over existing state-of-the-art methods like AlphaFold, OmegaFold and IgFold. However, the weakness of this paper is its novelty. The proposed workflow is almost the same as IgFold. In fact, it uses the same antibody language model and the same architecture as AlphaFold. The only difference is perhaps a better search of templates using foldseek that may explain the mode better performance. However, even foldseek is developed by another group.

**Summary Of The Paper:**

This paper proposes an antibody folding model based on existing architectures, including a protein language model (AntiBERTy), the EvoFormer and structure module from alphafold. The authors searched for templates from PDBs using Foldseek and extracted features using a similar workflow to AlphaFold. The method outperformed Alphafold and other folding algorithms based on protein language models in terms of RMSD (both framework and individual CDR regions).

**Summary Of The Review:**

Despite its empirical value as a better antibody folding tool, I believe this paper should be submitted to a biophysics journal and reviewed by more domain experts who can properly interpret the significance of the results. The algorithmic innovation of this paper is weak, thus I vote for rejection.

---

### Official Review · Reviewer_hjqN · 2022-11-02

**Confidence:** 3
**Correctness:** 3
**Technical Novelty And Significance:** 2
**Empirical Novelty And Significance:** 2
**Recommendation:** 6

**Clarity, Quality, Novelty And Reproducibility:**

Novelty

As already described, the novelty of the proposed work is limited compared to existing work, which the authors cite.

Quality

There are no theoretical results, so the technical quality of the work cannot be evaluated in those terms.

The experimental results seem fine. The results in Table 2 show that it is important to use an antibody language model, or else (as shown in xTrimoABFold-ESM), the results are not competitive with the state of the art.

It is nice to see that the proposed approach is effective for CDR3, but that is typically a T-cell response mechanism. Considering the aim of the study (i.e., antibody/B-cell responses), I’m not sure this is really relevant.

Reproducibility

The reproducibility of the work seems low. As far as I could tell, neither code nor datasets are provided. Considering the very technical and domain-specific contributions of this work, this lack of reproducibility seems like a limitation.

Clarity

The paper has numerous typos and needs another round of editing.

The reference format is inconsistent.

The paper is otherwise easy enough to follow.

**Strength And Weaknesses:**

The main strength of the paper is the empirical performance compared to existing state-of-the-art methods. In particular, the wall clock time comparison shows that the approach might meaningfully change what is feasible in terms of structure prediction and comparison.

The main weakness is the lack of novelty. The current work basically just plugs in existing approaches into an existing framework. I appreciate that choosing the right plug-ins isn’t easy, but the theoretical advances seem hard to pinpoint.


**Summary Of The Paper:**

In this work, the authors propose a set of changes to the AlphaFold2 architecture to improve its performance on antibody folding. Most importantly, they show how to replace the MSA component of AlphaFold2 with a much faster encoder approach. Consequently, the proposed approach achieves competitive or better performance while reducing the runtime by two orders of magnitude.


**Summary Of The Review:**

The paper shows that various engineering approaches indeed improve antibody folding prediction.

---

### Official Review · Reviewer_bKCt · 2022-11-03

**Confidence:** 2
**Clarity, Quality, Novelty And Reproducibility:** Please see above.
**Correctness:** 2
**Technical Novelty And Significance:** 2
**Empirical Novelty And Significance:** 2
**Recommendation:** 5

**Strength And Weaknesses:**

The two claimed strengths are:
- Antibody language model
- Fast multi-modal search technique.

However, I don't see any solid technical contribution in these. I am not actively working on this problem, so I cannot comment about the practical implications of the techniques.

Weakness:
- While the current method uses many deep learning techniques, the actual contribution seems to be entirely in the computational biology domain.
- The writing of the method is not friendly to a beginner. For example, it is not clear to me how the PLM in the subsequent model.
- Finally, the authors only compare with existing methods on the antibody dataset. I am not sure if the existing methods are specialized for the antibody structure prediction problem, which makes the comparison unfair.

**Summary Of The Paper:**

This is computational biology paper which reports a novel protein structure folding technique. The key idea is to improve the prediction of antibody protein structures using an antibody sequence model, which they call antibody language model. For this, the authors create a database of routine structures from the PDB. The authors then use recurrent neural networks similar to transformers for creating the antibody language model. The authors use the antibody language model along with the Alphafold2 to predict protein structures. The authors also claim to design a fast multi-modal sequence search technique.

The author show improvement on TMscore and RM is the over many recent baselines, on a database comprising of only antibody proteins.

**Summary Of The Review:**

In summary, I am not sure if ICLR is the right venue for the paper. While the paper uses some deep learning techniques, it's contributions are in the domain of computational biology. Another problem is that the document is not very accessible to someone not working on the particular problem of protein folding.

On the positive side, the authors show improvements over some very recent and impressive baselines.

---

### Official Review · Reviewer_pjvj · 2022-11-04

**Confidence:** 5
**Correctness:** 2
**Technical Novelty And Significance:** 1
**Empirical Novelty And Significance:** 1
**Recommendation:** 1

**Clarity, Quality, Novelty And Reproducibility:**

- Clarity: The paper itself is quite poorly written with many grammatical errors and typos, many of which could have been avoided with a simple copy-and-paste into a word-processor. The methods and results are also not well organized or explained. For example:
    - As mentioned before, there is very little background knowledge provided about antibodies.
    - The section "Template Searching Algorithms and Tools" is extremely difficult to read. There is very little background into
    - In equation (6), what exactly do you mean $T_j$ is an SE(3) transformation? What is the $j$ subscript indicating?
    - In equation (7), what is the purpose of $\epsilon$?
    - In equation (8), why divide by $Z$? What does $Z$ represent. Without the $1/Z$ factor, it looks like the average of $min(d_{clamp}, d_{ij})$.
    - Many comparisons are made against AlphaFold2. One of the most relevant hyperparameters to mention would then be MSA depth. However, there is no mention of this hyperparameter nor an ablation compared against this.
    - You make a mention of DeepAb and ABlooper but there is no comparison made with those methods.
    - In Figure 3, median time of what? There is no mention of what is actually being measured.
    - There should be some figure(s) that actually show the ground-truth and predicted 3D structures (for both xTrimoABFold and a comparable baseline).
- Quality: I'm quite unsure about the numbers for Table 2, particularly for IgFold. RMSD of 14 is extremely large, and the numbers for IgFold look more like the result of poor implementation. If this result is indeed valid, then the authors should include some exploration/explanation into why this may be so, as IgFold could be considered state-of-the-art. In Table 3, the RMSD values for the CDR3 region are almost too good to be true. In reality, from the perspective of antibody design and engineering, the CDR3 region is the most variable and difficult to model. The RMSD values computed for even the baselines is extremely low, which is quite surprising. This may, however, be due to the nature of the dataset. The authors specifically use single chains for the antibody dataset. This part requires further exploration.
- Novelty: The paper has very little novelty. The authors generate a filtered dataset from data that is publicly available (though it is not clear why the authors did not simply choose to use a dataset that others have used in the literature). The authors use the AlphaFold2 architecture (without MSAs) and the same loss function (without MSA loss). The authors add a fine-tuning loss that penalizes RMSD for CDRs.
- Reproducibility: The authors do not provide any open source code or the data they trained on.

**Strength And Weaknesses:**

Strengths:
- The results in Tables 2 and 3 are astounding, though some of the numbers may require further exploration/explanation.

Weaknesses:
- The method itself is not novel at all. The authors essentially reuse the loss function and architecture from AlphaFold2 (without MSAs).
- The analysis of the results are quite shallow. For example, a structure prediction paper with no figures with structures is quite strange and unusual.
- The results are almost too good to be true. In addition, the results for IgFold look confusingly poor. IgFold can be considered the current state-of-the-art.
- Please include more background information on antibodies and proteins as a whole. For example, why do we even care about CDRs? There was not a single mention in the paper about what the CDR actually is and what is so critical about the CDR to an antibody's function.
- Many grammatical errors and typos, to the point that the paper looks like it was rushed and ill-prepared. I understand that the authors may not be fluent in English, but many of these errors could easily have been fixed by copy-and-pasting into standard text editors/word processors with spelling and grammar checking.

**Summary Of The Paper:**

Antibody structure prediction is a highly sought after problem in industrial drug discovery pipelines. As with standard proteins, accurate predictions can lead to a better understanding of an antibody's function. Unfortunately, current methods for antibody structure prediction are not yet able to produce high-resolution predictions, particularly at the variable CDR regions of the antibodies. The paper presents xTrimoABFold, which is able to achieve high accuracy on a curated antibody structure dataset at much lower computational cost.

**Summary Of The Review:**

The paper is quite poorly written. Besides the numerous grammatical errors and typos, the content of the paper, as it currently stands, would not qualify for publication. There is very little background knowledge provided, e.g. why do we care about CDRs for antibodies and how does this make the problem very different than standard protein structure prediction. The method itself is unoriginal as it uses the exact same architecture (without MSAs) and loss function (again, without MSA contribution) as AlphaFold2. The authors use FoldSeek for template search. The analysis of the results is also quite shallow. I have never seen a biological structure-prediction paper without a figure that displays 3D structures. The results in Table 2 and 3, while impressive, look almost too impressive. In addition, the results for IgFold look almost unreal as IgFold could be considered the current state-of-the-art for antibody structure prediction.

---

### Decision · Program_Chairs · 2023-01-20

**Decision:**

Reject

**Justification For Why Not Higher Score:**

There are several unaddressed concerns.

**Justification For Why Not Lower Score:**

N/A

**Metareview: Summary, Strengths And Weaknesses:**

The paper proposes to accomplish Antibody structure prediction through pre-trained Antibody language model.
The results are very encouraging but there were several questions(namely the methods and datasets used for comparison, suitability of ICLR as venue, etc). Unfortunately the authors did not respond with a rebuttal. At this stage, given that the queries are not satisfactorily addressed, the paper is recommended for reject.


**Summary Of Ac-Reviewer Meeting:**

Not needed